# Post-remission outcomes in AML patients with high hyperleukocytosis and inaugural life-threatening complications

Sofiane Fodil[1,2], Sylvie Chevret[3], Camille Rouzaud[1], Sandrine Valade[4], Florence Rabian[5], Eric Mariotte[4], Emmanuel Raffoux[1], Raphael Itzykson[1], Nicolas Boissel[5], Marie Sébert[6,7], Lionel Adès[6,7], Lara Zafrani[4], Elie Azoulay[4], Etienne Lengliné[1] *

1 Hématologie Adulte, Hôpital Saint-Louis, APHP, Université Paris Diderot, Paris, France, 2 Sorbonne Université, Paris, France, 3 Service de Biostatistique et Information Médicale, Hôpital Saint-Louis, Paris, France, 4 Medical Intensive Care Unit, Hôpital Saint-Louis, APHP, Université Paris Diderot, Paris, France, 5 Hématologie Adolescents et Jeunes Adultes, Hôpital Saint-Louis, APHP, Université Paris Diderot, Paris, France, 6 Hématologie Seniors, Hôpital Saint-Louis, APHP, Université Paris Diderot, Paris, France, 7 Université de Paris and INSERM U944, Paris, France

* etienne.lengline@aphp.fr

## Abstract

### Introduction

Patients with hyperleukocytic (HL) acute myeloid leukemia (AML) are at higher risk of early death. Initial management of these patients is challenging, not fully codified and heterogenous. Retrospective studies showed that several symptomatic measures might decrease early death rate but long-term data are scarce. We aimed to analyze whether the therapeutic measures carried out urgently at diagnosis may influence the outcome among HL AML patients having achieved who survived inaugural complications.

### Methods

We retrospectively reviewed all medical charts from patients admitted to Saint-Louis Hospital between January, 1$^{st}$ 1997 and December, 31st 2018 with newly diagnosed AML and white blood cell (WBC) count above 50x10$^9$/L. Outcome measures were cumulative incidence of relapse (CIR), treatment-related mortality (TRM) defined as relapse-free death, and overall survival. Univariate and multivariate analyses were performed using Cox proportional hazards models.

### Results

A total of 184 patients with HL AML in complete remission (CR) were included in this study. At 2 years after CR. 62.5% of patients were alive, at 5 years, cumulated incidence of relapse was 55.8%. We found that every therapeutic measure, including life-sustaining therapies carried out in the initial phase of the disease, did not increase the relapse risk. The use of hydroxyurea for more than 4 days was associated with a higher risk of relapse. At the end of

**Data Availability Statement:** All relevant data are within the paper and its Supporting information files.

**Funding:** The author(s) received no specific funding for this work.

**Competing interests:** The authors have declared that no competing interests exist.

**Abbreviations:** Allogeneic HSCT, Allogeneic hematopoietic stem cell transplantation; AML, Acute myeloid leukemia; CR, complete remission; DIC, Disseminated intravascular coagulation; ELN, European LeukemiaNet; FLT3, FMS like tyrosine kinase; HL, Hyperleukocytic; HR, Hazard ratio; ICU, Intensive care unit; NGS, Next generation sequencing; RRT, Renal replacement therapy; TLS, tumor lysis syndrome; WBC, White blood cell.

the study, 94 patients (51.1%) were still alive including 23 patients out of 44 aged less than 60 yo that were able to return to work.

## Conclusion

We show that the use of emergency measures including life sustaining therapies does not come at the expense of a higher risk of relapse or mortality, except in the case of prolonged use of hydroxyurea. Patients with HL AML should be able to benefit from all available techniques, regardless of their initial severity.

## Introduction

Five to twenty percent of patients with Acute Myeloid Leukemia (AML) initially presents with high circulating blast cells, exceeding 50-100x$10^9$/L. In this subset of patients, hyperleukocytic AML (HL AML) is considered as a medical emergency as reported mortality in the first days after diagnosis are as high as 6% to 20% [1, 2]. Hyperleukocytosis is more frequently observed in monocytic AML (M4/M5 in the FAB classification) [3], in AML with MLL rearrangements [4] and in FMS like tyrosine kinase (FLT3) mutated/duplicated AML [5]. These patients are exposed to life-threatening complications responsible for multiple organ failures including cerebral or pulmonary leukostasis [6], solid organ infiltration [7], tumor lysis syndrome [8] or severe thrombosis or hemorrhages related to disseminated intravascular coagulation (DIC) [9].

Despite progress and improvement made to improve the short-term survival of these patients in the last decade [10], management remains extremely heterogeneous among countries, hospitals and even clinicians in charge of the patients, and is poorly codified [2]. Indeed, therapeutic strategies guidelines are mainly based on low-grade levels of evidence like expert opinions or retrospective cohort study. To date, no prospective randomized controlled trial has ever been conducted in this population. Nevertheless, Hyperleukocytic AML requires prompt actions to prevent rapid deterioration and early death [2]. Several reports described that 15 to 50% are managed in the intensive care unit (ICU) [11, 12] mainly for respiratory failure [13] or intensive monitoring [14, 15]. Those patients are also at risk of hemodynamic, neurologic or renal failure and require a high level of life-sustaining therapies resources: use of mechanical ventilation [16], catecholamines or renal replacement therapy [17]. Despite conflicting results, including a recent international retrospective study that failed to find a survival benefit, leukapheresis procedures are still used in some centers to rapidly clear peripheral circulating blast cells [18, 19]. On the other hand, Hydroxyurea is commonly administered to reduce the tumoral bulk before administration of intensive chemotherapy in order to lower the risk of dramatic tumor lysis syndrome (TLS) [20, 21]. Aggressive transfusions with various labile blood products such as packed red cells, platelets and fresh frozen plasma are widely used to prevent and treat or leukostasis consequences and DIC-associated bleeding [9]. Finally, corticosteroids are used to prevent and treat neurologic or pulmonary complication of leukostasis [22, 23].

Most of these early interventions have been implemented over time and are the subject of recommendations based on the evaluation of a short-term or very short-term benefit in a context of low methodological robustness of the studies conducted.

We therefore aimed to analyze whether the therapeutic measures carried out urgently at diagnosis may influence outcomes after complete remission (CR) in HL AML patients who survived inaugural complications.

## Materiel and methods

### Patients

All consecutive adults ($\geq$ 18 years old) patients admitted to our institution between January 1997 and December 2018 with a newly diagnosed AML according to WHO classification were retrospectively included if they 1/had an initial white blood cell (WBC) count exceeding $50 \times 10^9$/L, 2/Received at least one intensive induction course of chemotherapy and 3/obtained a complete remission as defined by 2010 European LeukemiaNet (ELN) AML Response Criteria [24]. Patients with acute promyelocytic leukemia and relapsed AML were excluded from the study. Datasets were anonymously extracted from medical charts and laboratory records. We collected clinical data at diagnosis and at the last visit for living patients, biological data on haematology, biochemistry and haemostasis and finally biological data on the characteristics of the leukaemia. For each patient, we also collected the treatments implemented as well as the different life sustaining therapies employed. Karyotype classification of the prognosis group was made according to the Medical Research Council (MRC) [25]. Risk stratification was performed using the 2010 ELN guidelines, due to lack of *FLT3*-ITD (Internal Tandem Duplication) allelic ratio and *TP53*, *ASXL1* and *RUNX1* mutational status for the oldest cases [24]. NPM1 and FLT3 status was performed by fragment analysis. The screening for mutation was carried out by next generation sequencing (NGS). The study was conducted in accordance with the Helsinki declaration. According to the French law on non-interventional study on health data, patients provided a non-opposition statement for anonymous data collection. This study was approved by the institutional review board of the French society of intensive care medicine (CE SRLF 21–88).

### Treatments and procedures

Our institution is a university hospital and a tertiary care center for blood diseases. During the study period, there was no formal guideline regarding pre-emptive ICU admission policy. ICU admission policy has not been modified over time, and there was no change in the proportion of patients with HL AML admitted during the study period. When a patient with HL AML was hospitalized in the ICU, both senior hematologists and intensivists managed the patient according to their respective competencies.

Supportive care and prevention of TLS were managed according to recommendations [26]. TLS was diagnosed according to international definition including hyperuricemia, hyperkaliemia, hyperphosphatemia, hypocalcemia and uremia [27]. DIC was diagnosed based on laboratory tests including platelets, fibrinogen level, D-dimer, prolonged Prothrombin Time (PT), according to international recommendations [28]. DIC management did not change over the study period and was based on the local guidelines that were subsequently published by international expert panel [28]. Platelets and fresh frozen plasma (15–30 mL/kg) transfusions were infused to maintain platelets over $50 \times 10^9$/L prothrombin time below 1.5N and fibrinogen over 1.5 g/L. Heparin was not used, except in case of thrombotic complications. Criteria used to diagnose leukostasis were respiratory symptom onset coinciding with a rapidly increasing WBC count and/or a WBC count $50 \times 10^9$/L in a patient not yet started on chemotherapy, symptoms of neurological leukostasis, the leukostasis score and improvement after chemotherapy initiation [16]. For patients in intensive care with organ failure, all organ replacement techniques were available 24/7 except ExtraCorpoeal Membrane Oxygenation (ECMO). Hydroxyurea could be started promptly at diagnosis for leukocytic reduction. Dexamethasone was added to induction chemotherapy in all patients who had a WBC count of at least $100 \times 10^9$/L or in patients with a WBC count over $50 \times 10^9$/L and clinical symptoms of leukostasis

until initial clinical improvement. Hydroxyurea and dexamethasone initiation was at the discretion of treating physician. Leukapheresis was not performed. Study patients received induction chemotherapy that included daunorubicin at a daily dose of 60–90 mg/m2 of body surface area daily for 3 days, or idarubicin at a daily dose of 8–12 mg/m2 daily for 5 days, together with a continuous intravenous infusion of cytarabine at a daily dose of 100–200 mg/m2 daily for 7 days [29]. CR and outcomes were defined according to ELN 2010 [24]. Patients who did not reach CR after the first course of chemotherapy could receive a salvage regimen if they had no uncontrolled infection or poor performance status. After achieving CR, consolidation chemotherapy or allogeneic hematopoietic stem cell transplantation (allogeneic HSCT) was performed, according to guidelines mainly based on age, comorbidities, oncogenetic classification and donor availability [24].

Karyotype classification of the prognosis group was made according to the Medical Research Council (MRC) [25]. Date of diagnosis was considered as the diagnostic bone marrow examination date and served to calculate survival and time to relapse. Relapse was defined as morphological leukemia recurrence after a first complete remission (molecular relapse was not considered).

## Statistical analysis

Descriptive statistics, that is, percentage for qualitative variables and median with interquartile range (IQR) for continuous variables, are reported. Patient characteristics were compared with the exact Fischer test for categorical variables and the Wilcoxon rank sum test for continuous ones.

Outcome measures were cumulative incidence of relapse (CIR), treatment-related mortality (TRM) defined as relapse-free death, and overall survival. All times were measured from the day of CR assessment. CIR and TRM were computed in a competing risk setting where both events compete with each other, and noninformative censoring at last follow-up. Overall survival was estimated by the Kaplan-Meier method.

For each endpoint, univariate and multivariate analyses were performed using Cox proportional hazards models. Multivariate Cox model included all the variables associated with the outcome at the 0.10 threshold. Full model was fitted on the complete cases only (n = 143) with model selection based on a stepwise procedure. A sensitivity analysis was performed by reproducing these models after multiple imputation by chained equations (MICE) on 30 complete datasets.

Point estimates of hazard ratio (HR) are reported together with 95% confidence intervals (CI). All tests were two-sided and p values of 0.05 or less were considered as statistically significant. Statistical analyses were performed using R 4.0.3 (https://www.R-project.org/).

## Results

### Population and emergency management

Between January 1997 and December 2018, 273 patients with hyperleukocytic AML were admitted to our hospital. Among them, 50 died within 40 days of the diagnosis before blood counts recovery and in 39 (14.2%) bone marrow evaluation did not meet CR criteria. A total of 184 patients with HL AML in CR were included in our study (Fig 1).

Baseline clinical and biological characteristics of the patients are presented in Table 1. 91 patients (49.5%) were men, median age was 47 IQR(32.5–60.5) years old. PS according to ECOG score was of 0 or 1 for 70.6% of patients. Regarding leukemia, median WBC count was 104 (71–158)x10$^9$/L at diagnosis with mostly monocytic differentiation (FAB M4/M5 AML) in 63.3%. Unfavorable karyotype was encountered in 27 patients (14.8%). NPM1 mutation was

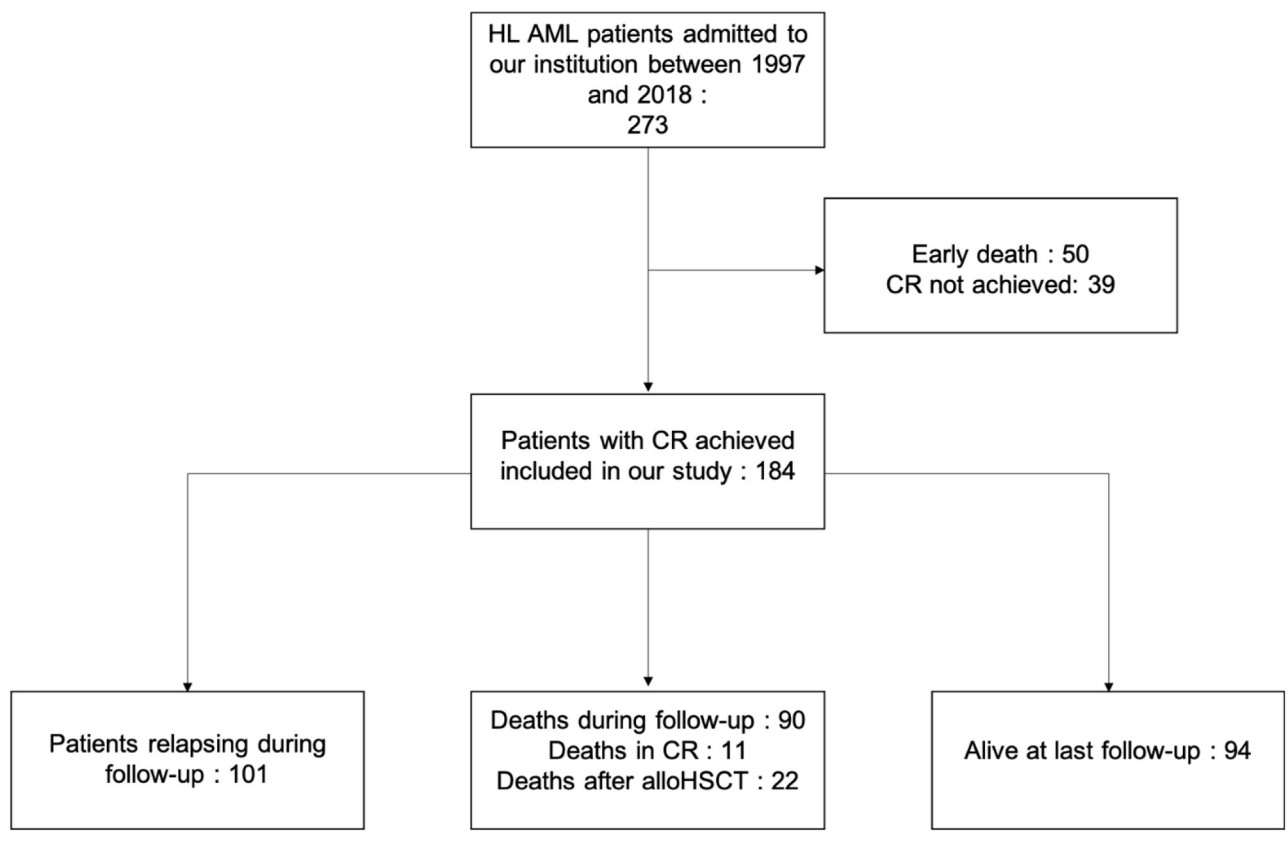

**Fig 1. Flow chart of patients study inclusion.**

tested in 141 patients and was found in 57 of them (40.3%). FLT3 abnormalities consisting of FLT3-ITD and FLT3-TKD mutations were found in 50 and 20 patients respectively out of 161 patients tested (42.9%).

Specific AML complications were observed in 112 patients (60.9%), including tumor lysis syndrome (TLS) in 84 patients (46.15%), Disseminated Intravascular Coagulation (DIC) in 42 patients (22.9%) and Leukostasis in 38 patients (20.6%).

All patients received induction chemotherapy with a standard treatment of "7+3" with anthracyclines (Daunorubicine or Idarubicine) and cytarabine. The median time from hospital admission to the initiation of chemotherapy was 4 days IQR(2–6). twenty-three patients received a third drug treatment which was mainly gemtuzumab-ozogamicin. CR assessment was assessed by cytology on bone marrow aspirate with a median time of 47 IQR(36–49) days after initiation of induction chemotherapy.

Initial management of patients included Hydroxyurea administration to 151 patients (82.5%) and dexamethasone to 69 patients (37.5%) corresponding to the 69 patients with clinical signs of leukostasis. Median time to administration from hospitalization of hydroxyurea and dexamethasone was 0 (0–1) day. ICU admission was decided for 88 patients (47.3%). sixty-four (34.7%) patients required organs supply during ICU stay, and it was oxygen administration for 58 patients (31.7%) and invasive mechanical ventilation in 20 patients (10.9%). Twenty patients (10.9%) received renal replacement therapy and 13 patients (7.1%) were treated with vasopressor. Among those patients, 7 patients (3.8%) had multi-organ failure (two or more organ failures) and required two or three life-sustaining therapies.

**Table 1. Baseline clinical and biological characteristics of included patients.**

| Parameters | | Baseline characteristics (IQR) |
|---|---|---|
| Age | | 47 (32.5–60.5) |
| Sexe: Male | | 91 (49.5%) |
| Weight (Kg) | | 70 (61.75–77.25) |
| PS | | 1 (1–2) |
| WBC count ($10^9$/L) | | 104 (71.85–158.5) |
| Platelets ($10^9$/L) | | 49 (24–97) |
| | | |
| Leucostasis | | 38 (20.6%) |
| TLS | | 84 (46.15%) |
| DIC | | 42 (22.95%) |
| De novo leukemia | | 163 (88.6%) |
| Secondary leukemia | | 21 (11.4%) |
| FAB M4/5 | | 112 (63.3%) |
| Karyotype | Favorable | 27 (14.8%) |
| | Intermediate | 128 (70.3%) |
| | Unfavorable | 27 (14.8%) |
| NPM1 mutation (N = 141) | | 57 (40.3%) |
| FLT3 mutation | FLT3-ITD (N = 168) | 51 (30.4%) |
| | FLT3-TKD (N = 161) | 20 (12.4%) |
| | | |
| Hydroxyurea | | 151 (82.5%) |
| Dexamethasone | | 69 (37.5%) |
| | | |
| ICU admission | | 88 (47.3%) |
| Oxygen administration | | 58 (31.7%) |
| Mechanical ventilation | | 20 (10.9%) |
| Acute Kidney Injury (GFR < 90mL/min) | | 104 (56.4%) |
| RRT | | 20 (10.9%) |
| Vasopressor therapy | | 13 (7.1%) |

## Relapse risk analysis

Median follow up was 25.9 IQR(13.1–60.8) months. Forty-two patients (22.8%) underwent allogeneic HSCT in first CR. During follow-up, 101 patients relapsed corresponding to an estimated 5-years cumulative incidence of relapse of 55.8% (95%CI, 48.1–62.8). Univariate and multivariate analyses were performed to disclose leukemia, patient, complication and emergency therapeutics factors associated with the risk of relapse (Table 2). As expected, an older age and an ELN 2010 molecular risk other than favorable were independently associated with relapse whereas allogeneic HSCT in first CR was associated with a reduction of this risk. In addition, Platelet count at baseline was also independently associated with relapse. The use of hydroxyurea was not statistically associated with the risk of relapse but we found that duration of hydroxyurea treatment (analyzed as a continuous variable) before intensive chemotherapy was statistically associated with the risk of relapse in both univariate and multivariate analysis (HR = 1.05 (1.01–1.1); p = 0.016). The observed effect was not related to the cytoreductive properties of hydroxyurea because neither baseline leukocytosis nor leukocytosis after hydroxyurea at the time of initiation of intensive chemotherapy was associated with the risk of

**Table 2. Univariate and multivariate analysis of relapse risk.**

| Parameters | HR (95%IC) | P-Value | HR (95%IC) | P-Value |
|---|---|---|---|---|
| | Univariate analysis | | Multivariate analysis | |
| **Age (years)** | 1.03 (1.02 to 1.04) | <0.0001 | 1.03 (1.02 to 1.05) | <0.0001 |
| **PS** | 1.08 (0.85; 1.38) | 0.54 | | |
| **Decade of inclusion** | | | | |
| [1997–2003] (N = 47) | 1.00 | | | |
| [2004–2009] (N = 46) | 1.02 (0.59; 1.75) | 0.95 | | |
| [2010–2015] (N = 52) | 0.74 (0.42; 1.28) | 0.27 | | |
| [2016–2019] (N = 39) | 1.12 (0.65; 1.95) | 0.68 | | |
| **Secondary AML** | 1.46 (0.83; 2.57) | 0.19 | | |
| **WBC baseline ($10^9$/L)** | 1.2 (0.79; 1.82) | 0.39 | | |
| **WBC on the day of induction ($10^9$/L)** | 1 (1; 1) | 0.47 | | |
| **Hemoglobin (d/dL)** | 1.04 (0.96; 1.12) | 0.39 | | |
| **Platelets ($10^9$/L)** | 1.01 (1; 1.02) | 0.032 | 1.40 (1.09 to 1.80) | 0.009 |
| | | | | |
| **Karyotype** | | | | |
| Favorable (N = 27) | 1.00 | | | |
| Intermediate (N = 128) | 2.09 (1.08; 4.05) | 0.029 | 0.69 (0.28 to 1.66) | 0.40 |
| Unfavorable (N = 27) | 2.15 (0.98; 4.75) | 0.058 | 0.70 (0.22 to 2.24) | 0.54 |
| **FLT3 abnormalities** | 1.62 (1.07; 2.44) | 0.022 | 1.54 (0.72 to 3.28) | 0.27 |
| **Allogeneic HSCT** | 0.46 (0.27; 0.81) | 0.006 | 0.28 (0.14 to 0.55) | 0.0002 |
| **ELN 2010 favorable** | 0.52 (0.32; 0.83) | 0.006 | 0.43 (0.26 to 0.71) | 0.001 |
| | | | | |
| **Hydroxyurea** | 1.48 (0.84; 2.6) | 0.18 | | |
| **Hydroxyurea duration (days)** | 1.05 (1.01; 1.1) | 0.016 | 1.05 (1.01 to 1.09) | 0.012 |
| **Dexamethasone** | 0.91 (0.6; 1.36) | 0.64 | | |
| **ICU admission** | 0.84 (0.57; 1.25) | 0.39 | | |
| | | | | |
| **Hypoxemia** | 0.77 (0.49; 1.19) | 0.24 | | |
| **Mechanical ventilation** | 0.59 (0.27; 1.27) | 0.18 | | |
| **Serum creatinine baseline (µmol/L)** | 1.01 (0.98; 1.04) | 0.69 | | |
| **Tumor Lysis Syndrome** | 0.95 (0.64; 1.41) | 0.81 | | |
| **RRT** | 0.87 (0.44; 1.72) | 0.68 | | |
| **RRT during chemotherapy** | 0.98 (0.16; 6.09) | 0.98 | | |
| **Catecholamines** | 0.22 (0.05; 0.88) | 0.032 | 0.17 (0.024 to 1.25) | 0.082 |
| **DIC** | 0.98 (0.6; 1.61) | 0.95 | | |

relapse (Table 2). Also, it was not an effect of the period of time as we did not find any effect of the decade of treatment on relapse risk.

Other emergency measures used at diagnosis were not associated with the risk of relapse. Neither the need for admission to intensive care (HR = 0.84 (0.57–1.25) p = 0.39) nor the need for life-sustaining therapy (mechanical ventilation: (HR = 0.59 (0.27–1.27) p = 0.18), renal replacement therapy: (HR = 0.87 (0.44;1.72) p = 0.68)) was significantly associated with an increased risk of relapse as shown in Fig 2. On the contrary, catecholamine use was significantly associated with less relapse (HR = 0.22 (0.05–0.88) p = 0.032) but performed only in a small subset of 13 patients. Finally, we did not find any correlation between the degree of organ failure at diagnosis and the subsequent risk of relapse after CR. The risk of relapse does not increase if the number of organ failures increases.

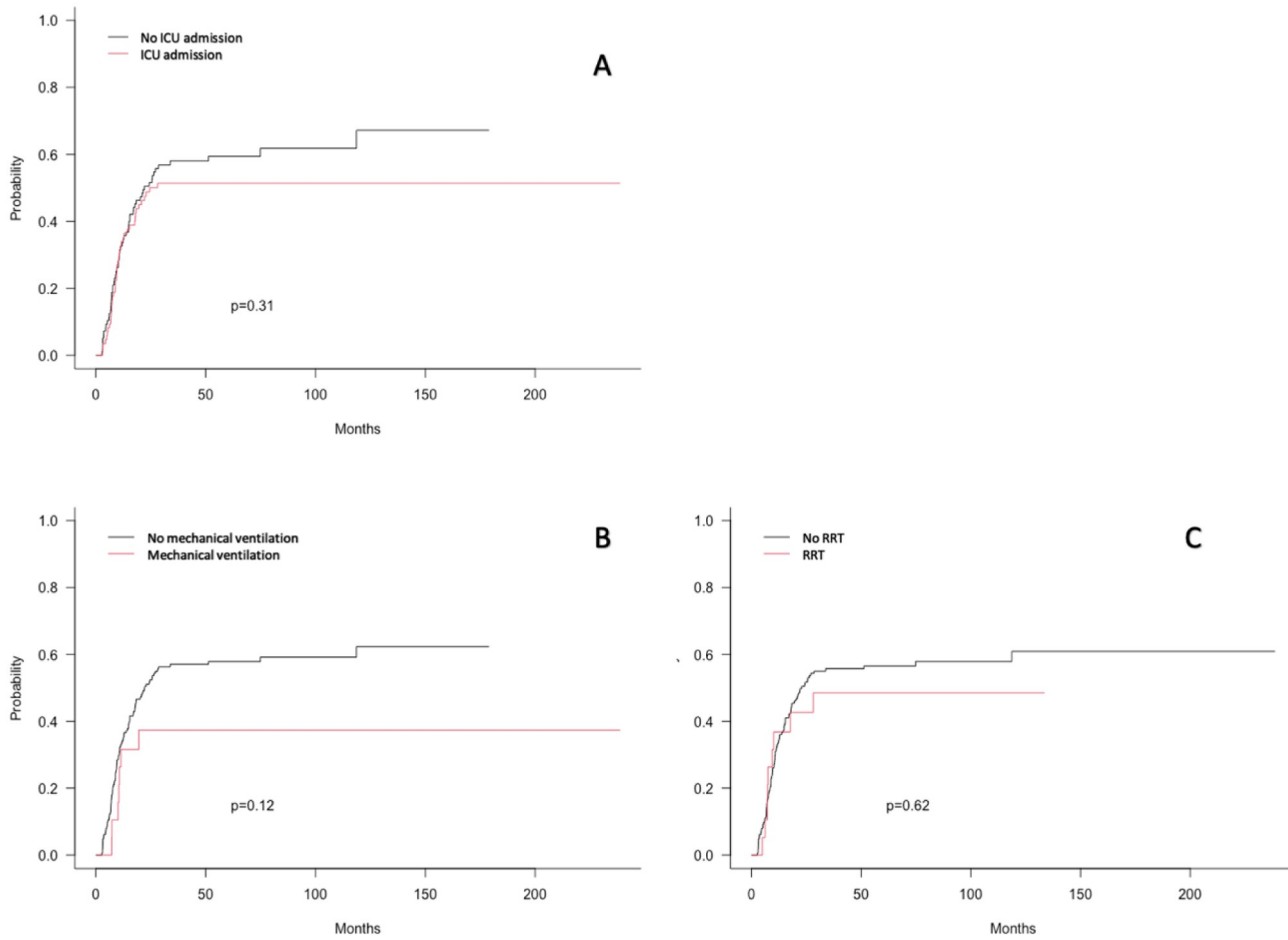

**Fig 2. Cumulative incidence of relapse.** (A): According to ICU admission status; (B): According to mechanical ventilation status; (C): According to renal replacement therapy status.

## Treatment-related mortality analysis

During the follow-up, 90 patients (48.9%) died, including 11 deaths in CR and 22 patients whom underwent allogeneic HSCT. Median time from CR to death was 15 IQR(4.5–41) months. In multivariate analysis, the risk of non relapse mortality was significantly associated with allogeneic HSCT (HR = 7.37 (1.49 to 36.5) (p = 0.015)) and the presence of an FLT3 mutation (Hazard Ratio of 5.36 (1.07 to 26.7) (p = 0.041) as shown in Table 3. Organ failures, inaugural complications as well as ICU admission, life-sustaining therapies, hydroxyurea or dexamethasone treatment were not associate with non-relapse mortality in univariate and multivariate analysis as shown in Fig 3.

## Overall survival

In this cohort, overall survival 2 years after CR achievement was 62.5%, (median not reached). Factors independently associated with death were an older age, high platelet count and genetic risk group, according to ELN 2010 (Table 4). The full range of emergency measures, including life sustaining therapies, performed in these patients does not affect overall long-term survival as shown in Table 4 and Fig 4.

**Table 3. Univariate and multivariate analysis of treatment-related mortality.**

| Parameters | HR (95%IC) | P-Value | HR (95%IC) | P-Value |
| --- | --- | --- | --- | --- |
| | Univariate analysis | | Multivariate analysis | |
| Age (years) | 0.99 (0.95; 1.02) | 0.48 | | |
| PS | 0.85 (0.39; 1.86) | 0.69 | | |
| Decade of inclusion | | | | |
| [1997–2003] (N = 47) | 1.00 | | | |
| [2004–2009] (N = 46) | 0.71 (0.12; 4.26) | 0.71 | | |
| [2010–2015] (N = 52) | 1.6 (0.4; 6.4) | 0.51 | | |
| [2016–2019] (N = 39) | 0 (0; Inf) | 1.00 | | |
| Secondary AML | 0.88 (0.11; 6.88) | 0.90 | | |
| WBC baseline ($10^9$/L) | 1 (0.29; 3.42) | 1.00 | | |
| WBC on the day of induction ($10^9$/L) | 1 (0.98; 1.01) | 0.41 | | |
| Hemoglobin (d/dL) | 0.88 (0.69; 1.13) | 0.31 | | |
| Platelets ($10^9$/L) | 1.02 (0.99; 1.05) | 0.11 | | |
| | | | | |
| Karyotype | | | | |
| Favorable (N = 27) | 0.34 (0.05; 2.44) | 0.29 | | |
| Intermediate (N = 128) | 0.42 (0.04; 4.68) | 0.48 | | |
| Unfavorable (N = 27) | 1.00 | | | |
| FLT3 abnormalities | 5.16 (1.04; 25.6) | 0.045 | 5.36 (1.07 to 26.7) | 0.041 |
| Allogeneic HSCT | 7.15 (1.9; 27) | 0.004 | 7.37 (1.49 to 36.5) | 0.015 |
| ELN 2010 favorable | 0.29 (0.06; 1.35) | 0.11 | | |
| | | | | |
| Hydroxyurea | 0.62 (0.16; 2.35) | 0.48 | | |
| Hydroxyurea duration (days) | 1.05 (0.92; 1.19) | 0.49 | | |
| Dexamethasone | 0.95 (0.28; 3.25) | 0.94 | | |
| ICU admission | 1.34 (0.41; 4.41) | 0.63 | | |
| | | | | |
| Oxygen administration | 1.19 (0.35; 4.08) | 0.78 | | |
| Mechanical ventilation | 1.74 (0.38; 8.06) | 0.48 | | |
| Serum creatinine baseline (µmol/L) | 1.04 (1; 1.09) | 0.049 | 0.09 (0.002 to 3.41) | 0.20 |
| DIC | 2.25 (0.66; 7.71) | 0.20 | | |

## Long term characteristics of survivors

At last follow-up, 94 patients (51.1%) were still alive and censored at a median of 61 months (29.4–93.3). We describe the characteristics of these patients at their last follow-up to understand the key elements of their health status (Table 5). At last follow-up, 69/72 (95.8%) patients had a performans status of 0 or 1. Among 44 patients under 60 y at diagnosis, 23 (52.3%) were able to return to work. The median time from diagnosis to return to work was 24.8 months. Among these survivor patients, 19 (20.4%) had previously experienced a relapse within a median of 20 months from diagnosis and 20 patients had undergone an allogeneic HSCT. As part of the pre-allograft assessment, patients undergo spirometry. Out of 23 patients who had performed a spirometry as a part of the pre-transplant assessment, it was abnormal in 11 (47.8%) of them.

## Discussion

In this study, we sought to analyze the possible post-remission influence, of measures commonly used to treat the inaugural complications of hyperleukocytic acute myeloid leukemias.

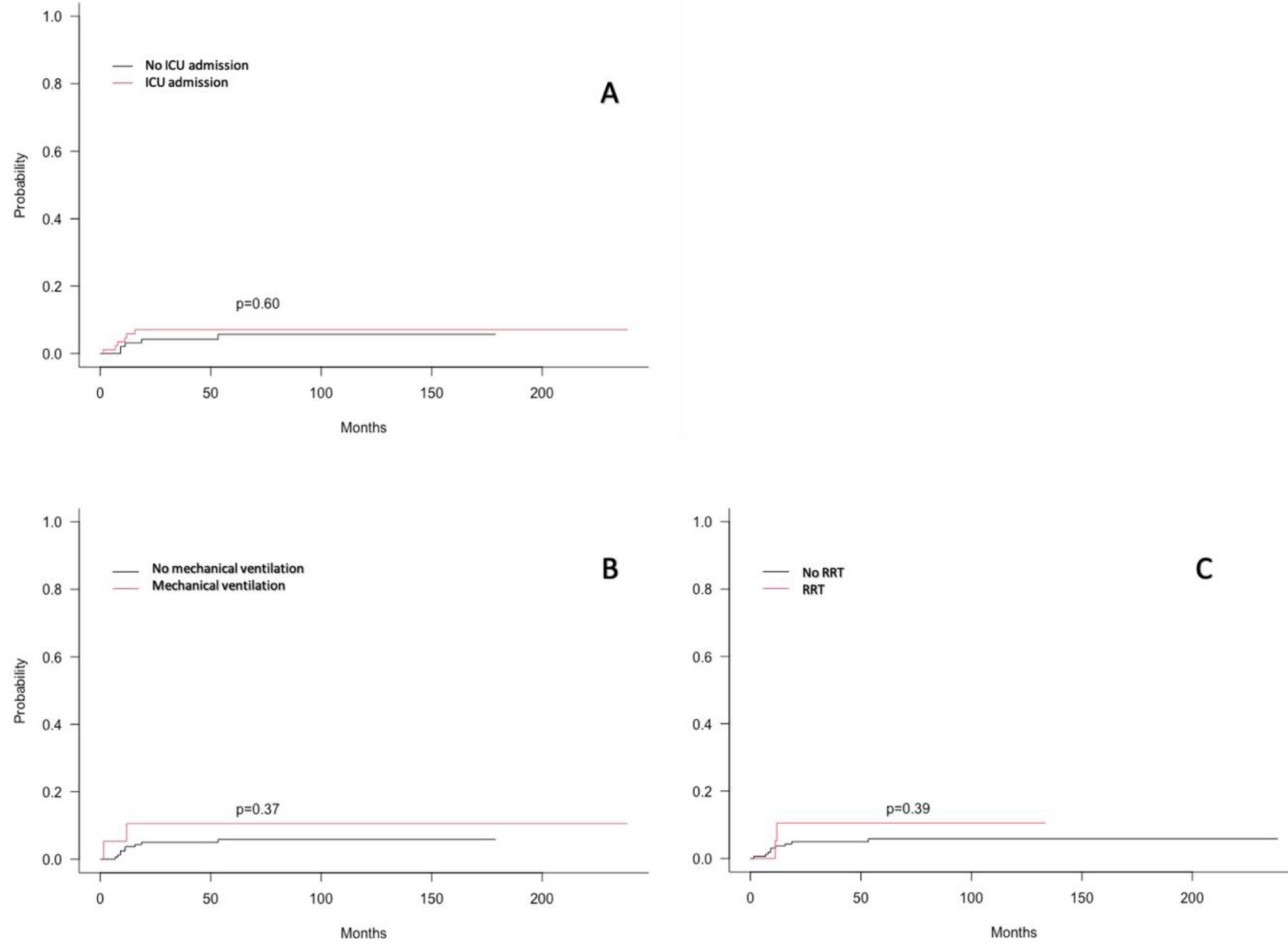

**Fig 3. Cumulative incidence of treatment-related mortality.** (A): According to ICU admission status; (B): According to mechanical ventilation status; (C): According to renal replacement therapy status.

We were able to analyze a large cohort of patients with AML and an initial leukocytosis greater than $50x10^9$/L who survived inaugural complications and achieved CR after intensive chemotherapy. These patients have, in real-life observational cohorts, a high incidence of inaugural life-threatening complications and are not well studied in prospective trials in the literature due to exclusion criteria and a selection bias towards less severe patients.

We showed reassuring evidence of no adverse impact of the use of life-sustaining techniques in these patients, including the use of renal replacement therapy (even at the same time as chemotherapy). Though, critically ill patients experience various significant physiological stress and alterations, related to organ ischemia, invasive procedures adverse events and prolonged immobilization leading to muscle loss, changes in drugs PK/PD and distribution volumes. Hepatic or renal failure may also modify the metabolism of AML chemotherapy with unknown consequences [30, 31]. To date, no chemotherapy pharmacokinetic data during dialysis is available [32]. Apart from one study suggesting that patients surviving an intensive care unit stay have a similar long-term outcome to others [11], the medical literature does not report any other data on the possible effect of the use of emergency resources and techniques on long-term outcome (in particular non-leukemia mortality and the risk of relapse after CR). Given the wide use of these techniques, it seems important to analyze their possible effect in

**Table 4. Univariate and multivariate analysis of deaths.**

| Parameters | HR (95%IC) | P-Value | HR (95%IC) | P-Value |
|---|---|---|---|---|
| | Univariate analysis | | Multivariate analysis | |
| Age | 1.02 (1.02; 1.03) | 0.002 | 1.03 (1.01; 1.05) | 0.001 |
| PS | 1.03 (0.79; 1.34) | 0.83 | | |
| Decade of inclusion | | | | |
| [1997–2003] (N = 47) | 1.00 | | | |
| [2004–2009] (N = 46) | 0.87 (0.46; 1.66) | 0.68 | | |
| [2010–2015] (N = 52) | 0.98 (0.52; 1.87) | 0.96 | | |
| [2016–2019] (N = 39) | 0.76 (0.41; 1.41) | 0.38 | | |
| Secondary AML | 1.29 (0.68; 2.42) | 0.44 | | |
| WBC baseline ($10^9$/L) | 1 (1; 1) | 0.31 | | |
| WBC on the day of induction ($10^9$/L) | 1 (0.99; 1) | 0.39 | | |
| Hemoglobin (d/dL) | 1.02 (0.94; 1.11) | 0.65 | | |
| Platelets ($10^9$/L) | 1.02 (1; 1.03) | 0.005 | 1.22 (1.07; 1.38) | |
| | | | | |
| Karyotype | | | | |
| Favorable (N = 27) | 1.00 | | | |
| Intermediate (N = 128) | 3.24 (1.4; 7.48) | 0.006 | 0.95 (0.33; 2.72) | 0.92 |
| Unfavorable (N = 27) | 4.19 (1.66; 10.56) | 0.002 | 1.34 (0.39; 4.65) | 0.65 |
| FLT3 abnormalities | 1.85 (1.19; 2.88) | 0.006 | 1.55 (0.75; 3.2) | 0.24 |
| Allogeneic HSCT | 1 (0.62; 1.53) | 0.99 | | |
| ELN 2010 favorable | 0.38 (0.22; 0.66) | 0.0006 | 0.46 (0.21; 0.99) | 0.047 |
| | | | | |
| Hydroxyurea | 1.1 (0.63; 1.93) | 0.73 | | |
| Hydroxyurea duration (days) | 1.04 (1; 1.08) | 0.077 | | |
| Dexamethasone | 0.99 (0.64; 1.53) | 0.97 | | |
| | | | | |
| Oxygen administration | 0.94 (0.59; 1.49) | 0.78 | | |
| Serum creatinine baseline (µmol/L) | 1.02 (1; 1.05) | 0.062 | | |
| DIC | 1.46 (0.9; 2.37) | 0.12 | | |

patients after CR has been achieved. These are therefore important arguments for not being reluctant to use all the available techniques that can currently be considered as treatments for inaugural complications and vital distress. One previous study showed that overall survival and leukemia-free survival of patients analyzed with a landmark at Day-30 of diagnosis did not differ between patients who stayed in ICU versus others [11]. This study did not focus on a population at particular risk of life-threatening distress and did not determine the potential effect of each life-support technique. Performance of a consolidation with allogeneic HSCT is the only determinant associated with long-term non-relapse mortality, indicating already known toxicity but it is important to note that aggressive supportive care like ICU admission or life-sustaining therapies (mechanical ventilation, renal replacement therapy (RRT) or the use of vasopressors) are not associated with therapy-related mortality indicating that those patient benefits from extensive supportive care.

We showed that classical factors related to leukemia (ELN genetic group) and to the patient (age) are, as expected, strongly associated with the risk of relapse and overall survival also in this specific population.

We find that the duration of treatment with hydroxyurea was associated with a risk of relapse. This drug is commonly considered for reducing early mortality but it could be at the

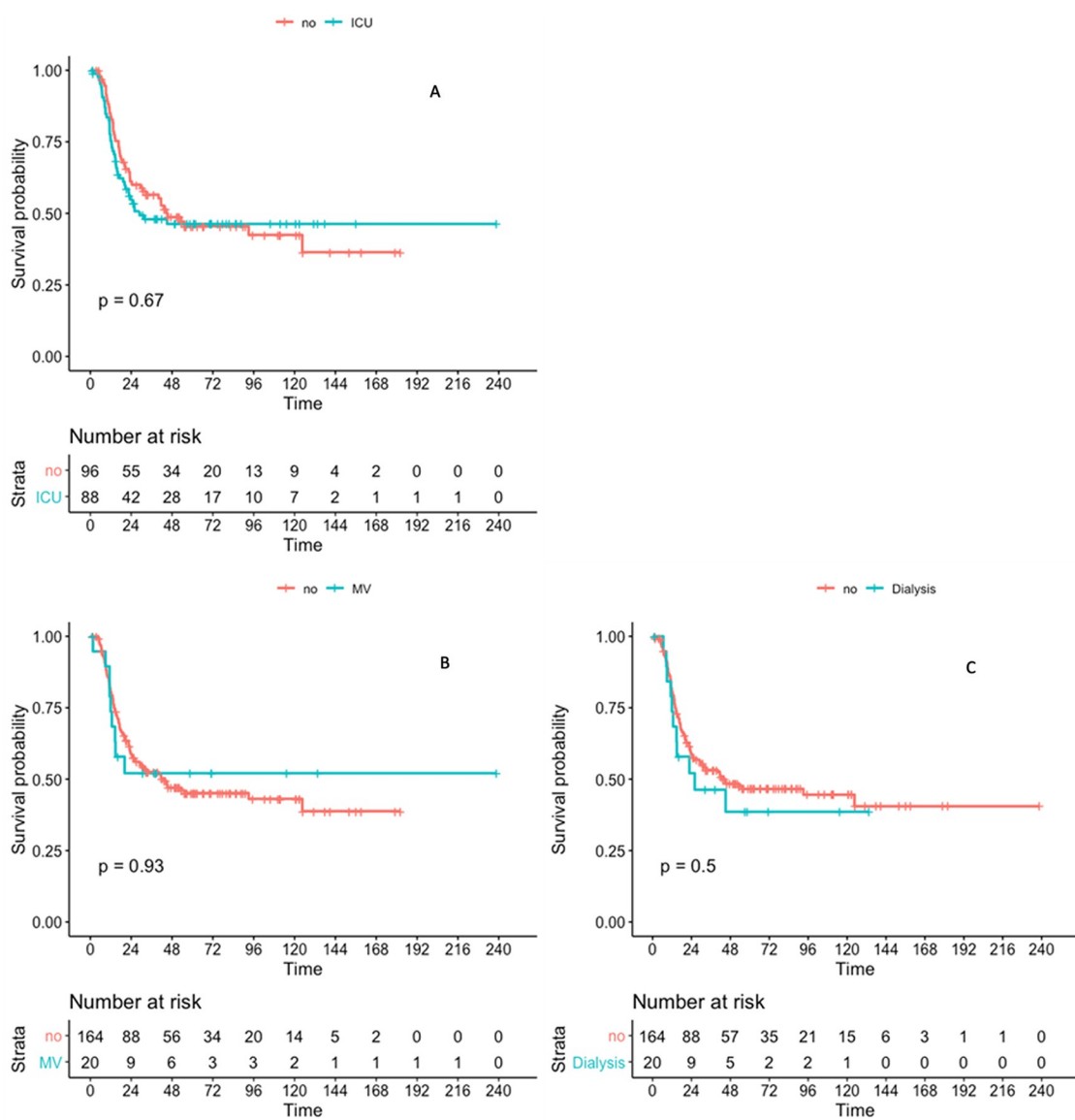

**Fig 4. Kaplan-Meyer analysis of overall survival.** (A): According to ICU admission status; (B): According to mechanical ventilation status; (C): According to renal replacement therapy status.

expense of higher rate of relapse. Given that hydroxyurea is usually given at the time of diagnosis until the start of intensive chemotherapy, it is not possible to know whether this is a specific effect of hydroxyurea or of or delayed administration of chemotherapy, or of another confounding factor which requires further analysis before confirming this result. Other cytoreduction strategies described in the literature include leukapheresis and high dose cyclophosphamide. Leukapheresis was the subject of a multicentric retrospective study but showed no beneficial effects on mortality [18, 19]. High dose cyclophosphamide has been shown to reduce early mortality but its long-term effects have not been studied [33]. Nonetheless, there are no strategies that have been shown to be superior to others and there is therefore a need for further study in this field.

**Table 5. Clinical and biological characteristics of patients alive.** N = 94.

| Parameters | At diagnosis | Last follow-up |
|---|---|---|
| PS | 1 (1–2) (n = 91) | 0 (0–1) (n = 72) |
| Return to work (n = 44) | | 23 (52.2%) |
| Time diagnosis-return to work (months) | | 24.4 (19.1–49) |
| Allograft (n = 93) | | 20 (21.5%) |
| Alive relapse free (n = 93) | | 74 (79.6%) |
| Time diagnosis-relapse (months) | | 20 (14.8–26.6) |
| Normal spirometry (n = 22) | | 11 (50%) |
| | | |
| Serum creatinine (μmol/L) (n = 68) | 83 (70–102.5) | 72 (62–93) |
| Hemoglobin (g/dL) | 8.7 (7.5–10.8) | 14 (12.7–14.4) |
| Platelets ($10^9$/L) | 40 (22–87) | 207 (150–256) |
| Ferritin (μg/L) | | 964 (449–1605) |

Bertoli et al. showed that the use of Dexamethasone was associated with a reduction in relapse risk but we did not find this association in our cohort although the doses of dexamethasone are similar between the two studies. The role of dexamethasone on short-term and long-term outcomes could be dissociated. The results are contradictory with some studies reporting a benefit on early mortality and some showing no benefit. In opposition, a benefit on long-term overall survival and on the risk of relapse is highlighted, which we do not find here. More studies on the effect and the pharmacodynamic of glucocorticoids are needed [22, 23].

The quality of life of the surviving patients appears to be good according to the PS according to very scarce data that we could retrieve. This is further verified by the fact that half of the patients under 60 years of age were able to resume their professional activity. This is an important fact compared to the initial severity of their disease. This study is the first study to focus on the long-term outcome of AML patients with hyperleukocytosis achieving CR. Some descriptive cohort studies of patients with hyperleukocytic AML describe the same range of relapse risk, therapy-related mortality and overall survival even when using different emergency therapies such as leukapheresis [19, 34, 35]. Nevertheless, our study has many limitations. This is a single center retrospective study and our findings should be confirmed in multi-center setting in order to avoid the effect of the heterogeneity of practices. In addition, patients at risk or with initial complications are classically excluded from large prospective multicenter therapeutic trials, which precludes the use of these data to draw conclusions on long-term outcomes in this subgroup of patients.

However, this is one of the only studies that focus on the impact of emergency treatment on the long-term outcome of patients with HL AML and It is very hard to implement prospective randomized clinical trials in this population. Also, we included patients over a very long period of time and we cannot be sure that there was no influence on analysis but we have verified that the treatment period (analyzed by decades) had no influence in univariate analysis on the different outcomes analyzed.

In conclusion, patients with HL AML continue to benefit in the long term from emergency measures taken at diagnosis, including replacement of vital organ failure and must continue to be managed using all available resources. Additional studies are required about cytoreduction strategies and acute complications management without forgetting their potential long-term effect on relapse or survival.

## Supporting information

**S1 Rawdata. Rawdata_longtermAML.**
(CSV)

## Author Contributions

**Conceptualization:** Etienne Lengliné.

**Data curation:** Sofiane Fodil.

**Formal analysis:** Sylvie Chevret.

**Methodology:** Sylvie Chevret, Etienne Lengliné.

**Supervision:** Elie Azoulay.

**Validation:** Camille Rouzaud, Sandrine Valade, Florence Rabian, Eric Mariotte, Emmanuel Raffoux, Raphael Itzykson, Nicolas Boissel, Marie Sébert, Lionel Adès, Lara Zafrani, Elie Azoulay, Etienne Lengliné.

**Writing – original draft:** Sofiane Fodil.

**Writing – review & editing:** Sofiane Fodil, Lionel Adès, Elie Azoulay, Etienne Lengliné.

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
