## [Decision Letter · Decision Letter 0]

9 May 2022

PONE-D-22-05709Post-remission outcomes in AML patients with high hyperleukocytosis and inaugural life-threatening complications.PLOS ONE

Dear Dr. Fodil,

Thank you for submitting your manuscript to PLOS ONE. After careful consideration, we feel that it has merit but after minor modificationsIndicate which changes you require for acceptance versus which changes you recommendAddress any conflicts between the reviews so that it's clear which advice the authors should followProvide specific feedback from your evaluation of the manuscriptPlease ensure that your decision is justified r >

We look forward to receiving your revised manuscript.

Kind regards,

Mohamed A Yassin, MD

Academic Editor

PLOS ONE

Journal Requirements:

3. Please delete the separate files, leaving only the tables in the manuscript file

4. Please include a copy of Table 5 which you refer to in your text on page 28.

Reviewers' comments:

Reviewer's Responses to Questions

**Comments to the Author**

1. Is the manuscript technically sound, and do the data support the conclusions?

Reviewer #1: Yes

Reviewer #2: Yes

2. Has the statistical analysis been performed appropriately and rigorously? 

Reviewer #1: Yes

Reviewer #2: Yes

3. Have the authors made all data underlying the findings in their manuscript fully available?

Reviewer #1: Yes

Reviewer #2: Yes

4. Is the manuscript presented in an intelligible fashion and written in standard English?

Reviewer #1: No

Reviewer #2: Yes

5. Review Comments to the Author

Reviewer #1: In the manuscript "Post-remission outcomes in AML patients with high hyperleukocytosis and inaugural life-threatening complications“ the authors investigated a large retrospective cohort of patients with AML presenting initially with hyperleucocytosis. The study is clinically relevant and the statics are performed correctly. However, there is still a need for thorough language editing.

Reviewer #2: The manuscript is clear, concise and well written. I recommend this manuscript be accepted.

Minor comments

-In the abstract part of the paper, please include about the statistical analysis

-In the method part of the document, you have to provide the data collection procedures. Please also define your study variables (dependent vs independent variable) clearly.

-Some references are too old (before 2000) and need to be updated

6. PLOS authors have the option to publish the peer review history of their article (what does this mean?). If published, this will include your full peer review and any attached files.

Reviewer #1: No

Reviewer #2: No

---

## [Author Response · Author response to Decision Letter 0]

30 May 2022

Response to Edits requested on your revised submission

1. Your ethics statement should only appear in the Methods section of your manuscript. If your ethics statement is written in any section besides the Methods, please move it to the Methods section and delete it from any other section. Please ensure that your ethics statement is included in your manuscript, as the ethics statement entered into the online submission form will not be published alongside your manuscript.

It has been changed according to your comments

2. 

Please ensure that you refer to Table 5 in your text as, if accepted, production will need this reference to link the reader to the Table.

It has been changed according to your comments

3. 

In line with our goal of ensuring long-term data availability to all interested researchers, PLOS’ Data Policy states that authors cannot be the sole named individuals responsible for ensuring data access (http://journals.plos.org/plosone/s/data-availability#loc-acceptable-data-sharing-methods).

We provided the raw data of the trial as requested

We thank the editor and 3 reviewers for their constructive comments. 

- Journal Requirements:

We checked and made sure that our manuscript meets PLOS ONE’s style requirements

We clarified this point and add a sentence to the Methods section “According to the French law on non-interventional study on health data, patients provided a written non-opposition statement for anonymous data collection. This study was approved by the institutional review board of the French society of intensive care medicine (CE SRLF 21-88).”

3. Please delete the separate files, leaving only the tables in the manuscript file

The changes were made according to the editor's comments

4. Please include a copy of Table 5 which you refer to in your text on page 28.

The changes were made according to the editor's comments

We changed the data availability statements and specified that our data will be only available upon direct request to the corresponding author

Comments to the Author

1. Is the manuscript technically sound, and do the data support the conclusions?

Reviewer #1: Yes

Reviewer #2: Yes

2. Has the statistical analysis been performed appropriately and rigorously?

Reviewer #1: Yes

Reviewer #2: Yes

3. Have the authors made all data underlying the findings in their manuscript fully available?

Reviewer #1: Yes

Reviewer #2: Yes

4. Is the manuscript presented in an intelligible fashion and written in standard English?

We would like to thank the reviewers once again for their time in reading and comments on our manuscript

Reviewer #1: No. Corrections have been made accordingly in the manuscript

Reviewer #2: Yes

5. Review Comments to the Author

Reviewer #1: In the manuscript "Post-remission outcomes in AML patients with high hyperleukocytosis and inaugural life-threatening complications“ the authors investigated a large retrospective cohort of patients with AML presenting initially with hyperleucocytosis. The study is clinically relevant and the statics are performed correctly. However, there is still a need for thorough language editing.

We thank the reviewer 1 for his comments, language editing has been made according to reviewer 1 comment

Reviewer #2: The manuscript is clear, concise and well written. I recommend this manuscript be accepted.

Minor comments

-In the abstract part of the paper, please include about the statistical analysis

We thank the reviewer 2 for his comments.

We included a sentence about the statistical analysis in the abstract. 

-In the method part of the document, you have to provide the data collection procedures. Please also define your study variables (dependent vs independent variable) clearly.

We clarified on how and which the data were collected in the methods part of the manuscript : “Datasets were anonymously extracted from medical charts and laboratory records. We collected clinical data at diagnosis and at the last visit for living patients, biological data on haematology, biochemistry and haemostasis and finally biological data on the characteristics of the leukaemia. For each patient, we also collected the treatments implemented as well as the different life sustaining therapies employed.”

-Some references are too old (before 2000) and need to be updated

Modifications has been made, according to the reviewer comment

6. PLOS authors have the option to publish the peer review history of their article (what does this mean?). If published, this will include your full peer review and any attached files.

Do you want your identity to be public for this peer review? For information about this choice, including consent withdrawal, please see our Privacy Policy.

Reviewer #1: No

Reviewer #2: No

---

## [Editor Report · Decision Letter 1]

17 Jun 2022

Post-remission outcomes in AML patients with high hyperleukocytosis and inaugural life-threatening complications.

PONE-D-22-05709R1

Dear Dr. Fodil,

We’re pleased to inform you that your manuscript has been judged scientifically suitable for publication and will be formally accepted for publication once it meets all outstanding technical requirements.

Kind regards,

Mohamed A Yassin, MD

Academic Editor

PLOS ONE
---

## [Editor Report · Acceptance letter]

27 Jun 2022

PONE-D-22-05709R1 

Post-remission outcomes in AML patients with high hyperleukocytosis and inaugural life-threatening complications. 

Dear Dr. Fodil:

I'm pleased to inform you that your manuscript has been deemed suitable for publication in PLOS ONE. Congratulations! Your manuscript is now with our production department. 

Kind regards, 

on behalf of

Dr. Mohamed A Yassin 

Academic Editor

PLOS ONE